# Generalized Cross Entropy Loss for Training Deep Neural Networks with Noisy Labels

**Zhilu Zhang**      **Mert R. Sabuncu**
Electrical and Computer Engineering
Meinig School of Biomedical Engineering
Cornell University
zz452@cornell.edu, msabuncu@cornell.edu

## Abstract

Deep neural networks (DNNs) have achieved tremendous success in a variety of applications across many disciplines. Yet, their superior performance comes with the expensive cost of requiring correctly annotated large-scale datasets. Moreover, due to DNNs' rich capacity, errors in training labels can hamper performance. To combat this problem, mean absolute error (MAE) has recently been proposed as a noise-robust alternative to the commonly-used categorical cross entropy (CCE) loss. However, as we show in this paper, MAE can perform poorly with DNNs and challenging datasets. Here, we present a theoretically grounded set of noise-robust loss functions that can be seen as a generalization of MAE and CCE. Proposed loss functions can be readily applied with any existing DNN architecture and algorithm, while yielding good performance in a wide range of noisy label scenarios. We report results from experiments conducted with CIFAR-10, CIFAR-100 and FASHION-MNIST datasets and synthetically generated noisy labels.

## 1   Introduction

The resurrection of neural networks in recent years, together with the recent emergence of large scale datasets, has enabled super-human performance on many classification tasks [21, 28, 30]. However, supervised DNNs often require a large number of training samples to achieve a high level of performance. For instance, the ImageNet dataset [6] has 3.2 million hand-annotated images. Although crowdsourcing platforms like Amazon Mechanical Turk have made large-scale annotation possible, some error during the labeling process is often inevitable, and mislabeled samples can impair the performance of models trained on these data. Indeed, the sheer capacity of DNNs to memorize massive data with completely randomly assigned labels [42] proves their susceptibility to overfitting when trained with noisy labels. Hence, an algorithm that is robust against noisy labels for DNNs is needed to resolve the potential problem. Furthermore, when examples are cheap and accurate annotations are expensive, it can be more beneficial to have datasets with more but noisier labels than less but more accurate labels [18].

Classification with noisy labels is a widely studied topic [8]. Yet, relatively little attention is given to directly formulating a noise-robust loss function in the context of DNNs. Our work is motivated by Ghosh et al. [9] who theoretically showed that mean absolute error (MAE) can be robust against noisy labels under certain assumptions. However, as we demonstrate below, the robustness of MAE can concurrently cause increased difficulty in training, and lead to performance drop. This limitation is particularly evident when using DNNs on complicated datasets. To combat this drawback, we advocate the use of a more general class of noise-robust loss functions, which encompass both MAE and CCE. Compared to previous methods for DNNs, which often involve extra steps and algorithmic modifications, changing only the loss function requires minimal intervention to existing architectures

and algorithms, and thus can be promptly applied. Furthermore, unlike most existing methods, the proposed loss functions work for *both* closed-set and open-set noisy labels [40]. Open-set refers to the situation where samples associated with erroneous labels do not always belong to a ground truth class contained within the set of known classes in the training data. Conversely, closed-set means that all labels (erroneous and correct) come from a known set of labels present in the dataset.

The main contributions of this paper are two-fold. First, we propose a novel generalization of CCE and present a theoretical analysis of proposed loss functions in the context of noisy labels. And second, we report a thorough empirical evaluation of the proposed loss functions using CIFAR-10, CIFAR-100 and FASHION-MNIST datasets, and demonstrate significant improvement in terms of classification accuracy over the baselines of MAE and CCE, under both closed-set and open-set noisy labels.

The rest of the paper is organized as follows. Section 2 discusses existing approaches to the problem. Section 3 introduces our noise-robust loss functions. Section 4 presents and analyzes the experiments and result. Finally, section 5 concludes our paper.

## 2   Related Work

Numerous methods have been proposed for learning with noisy labels with DNNs in recent years. Here, we briefly review the relevant literature. Firstly, Sukhbaatar and Fergus [35] proposed accounting for noisy labels with a confusion matrix so that the cross entropy loss becomes

$$\mathcal{L}(\theta) = \frac{1}{N}\sum_{n=1}^{N} -\log p(\widetilde{y} = \widetilde{y}_n | \boldsymbol{x}_n, \theta) = \frac{1}{N}\sum_{n=1}^{N} -\log(\sum_{i}^{c} p(\widetilde{y} = \widetilde{y}_n | y = i)p(y = i | \boldsymbol{x}_n, \theta)), \quad (1)$$

where $c$ represents number of classes, $\widetilde{y}$ represents noisy labels, $y$ represents the latent true labels and $p(\widetilde{y} = \widetilde{y}_n | y = i)$ is the $(\widetilde{y}_n, i)$'th component of the confusion matrix. Usually, the real confusion matrix is unknown. Several methods have been proposed to estimate it [11, 14, 32, 17, 12]. Yet, accurate estimations can be hard to obtain. Even with the real confusion matrix, training with the above loss function might be suboptimal for DNNs. Assuming (1) a DNN with enough capacity to memorize the training set, and (2) a confusion matrix that is diagonally dominant, minimizing the cross entropy with confusion matrix is equivalent to minimizing the original CCE loss. This is because the right hand side of Eq. 1 is minimized when $p(y = i | \boldsymbol{x}_n, \theta) = 1$ for $i = \widetilde{y}_n$ and 0 otherwise, $\forall n$.

In the context of support vector machines, several theoretically motivated noise-robust loss functions like the ramp loss, the unhinged loss and the savage loss have been introduced [5, 38, 27]. More generally, Natarajan et al. [29] presented a way to modify any given surrogate loss function for binary classification to achieve noise-robustness. However, little attention is given to alternative noise robust loss functions for DNNs. Ghosh et al. [10, 9] proved and empirically demonstrated that MAE is robust against noisy labels. This paper can be seen as an extension and generalization of their work.

Another popular approach attempts at cleaning up noisy labels. Veit et al. [39] suggested using a label cleaning network in parallel with a classification network to achieve more noise-robust prediction. However, their method requires a small set of clean labels. Alternatively, one could gradually replace noisy labels by neural network predictions [33, 36]. Rather than using predictions for training, Northcutt et al. [31] offered to prune the correct samples based on softmax outputs. As we demonstrate below, this is similar to one of our approaches. Instead of pruning the dataset once, our algorithm iteratively prunes the dataset while training until convergence.

Other approaches include treating the true labels as a latent variable and the noisy labels as an observed variable so that EM-like algorithms can be used to learn true label distribution of the dataset [41, 18, 37]. Techniques to re-weight confident samples have also been proposed. Jiang et al. [16] used a LSTM network on top of a classification model to learn the optimal weights on each sample, while Ren, et al. [34] used a small clean dataset and put more weights on noisy samples which have gradients closer to that of the clean dataset. In the context of binary classification, Liu et al. [24] derived an optimal importance weighting scheme for noise-robust classification. Our method can also be viewed as re-weighting individual samples; instead of explicitly obtaining weights, we use the softmax outputs at each iteration as the weightings. Lastly, Azadi et al. [2] proposed a regularizer that encourages the model to select reliable samples for noise-robustness. Another method that uses

knowledge distillation for noisy labels has also been proposed [23]. Both of these methods also require a smaller clean dataset to work.

# 3 Generalized Cross Entropy Loss for Noise-Robust Classifications

## 3.1 Preliminaries

We consider the problem of k-class classification. Let $\mathcal{X} \subset \mathbb{R}^d$ be the feature space and $\mathcal{Y} = \{1, \cdots, c\}$ be the label space. In an ideal scenario, we are given a clean dataset $D = \{(\boldsymbol{x}_i, y_i)\}_{i=1}^n$, where each $(\boldsymbol{x}_i, y_i) \in (\mathcal{X} \times \mathcal{Y})$. A classifier is a function that maps input feature space to the label space $f : \mathcal{X} \to \mathbb{R}^c$. In this paper, we consider the common case where the function is a DNN with the softmax output layer. For any loss function $\mathcal{L}$, the (empirical) risk of the classifier $f$ is defined as $R_{\mathcal{L}}(f) = \mathbb{E}_D[\mathcal{L}(f(\boldsymbol{x}), y_{\boldsymbol{x}})]$, where the expectation is over the empirical distribution. The most commonly used loss for classification is cross entropy. In this case, the risk becomes:

$$R_{\mathcal{L}}(f) = \mathbb{E}_D[\mathcal{L}(f(\boldsymbol{x}; \boldsymbol{\theta}), y_{\boldsymbol{x}})] = -\frac{1}{n} \sum_{i=1}^n \sum_{j=1}^c \boldsymbol{y}_{ij} \log f_j(\boldsymbol{x}_i; \boldsymbol{\theta}), \qquad (2)$$

where $\boldsymbol{\theta}$ is the set of parameters of the classifier, $\boldsymbol{y}_{ij}$ corresponds to the $j$'th element of one-hot encoded label of the sample $\boldsymbol{x}_i$, $\boldsymbol{y}_i = \boldsymbol{e}_{y_i} \in \{0,1\}^c$ such that $\mathbf{1}^\top \boldsymbol{y}_i = 1 \ \forall \ i$, and $f_j$ denotes the $j$'th element of $f$. Note that, $\sum_{j=1}^n f_j(\boldsymbol{x}_i; \boldsymbol{\theta}) = 1$, and $f_j(\boldsymbol{x}_i; \boldsymbol{\theta}) \geq 0, \forall j, i, \boldsymbol{\theta}$, since the output layer is a softmax. The parameters of DNN can be optimized with empirical risk minimization.

We denote a dataset with label noise by $D_\eta = \{(\boldsymbol{x}_i, \widetilde{y}_i)\}_{i=1}^n$ where $\widetilde{y}_i$'s are the noisy labels with respect to each sample such that $p(\widetilde{y}_i = k|y_i = j, \boldsymbol{x}_i) = \eta_{jk}^{(\boldsymbol{x}_i)}$. In this paper, we make the common assumption that noise is conditionally independent of inputs given the true labels so that

$$p(\widetilde{y}_i = k|y_i = j, \boldsymbol{x}_i) = p(\widetilde{y}_i = k|y_i = j) = \eta_{jk}.$$

In general, this noise is defined to be *class dependent*. Noise is *uniform* with noise rate $\eta$, if $\eta_{jk} = 1 - \eta$ for $j = k$, and $\eta_{jk} = \frac{\eta}{c-1}$ for $j \neq k$. The risk of classifier with respect to noisy dataset is then defined as $R_{\mathcal{L}}^\eta(f) = \mathbb{E}_{D_\eta}[\mathcal{L}(f(\boldsymbol{x}), \widetilde{y}_{\boldsymbol{x}})]$.

Let $f^*$ be the global minimizer of the risk $R_{\mathcal{L}}(f)$. Then, the empirical risk minimization under loss function $\mathcal{L}$ is defined to be *noise tolerant* [26] if $f^*$ is a global minimum of the noisy risk $R_{\mathcal{L}}^\eta(f)$.

A loss function is called *symmetric* if, for some constant $C$,

$$\sum_{j=1}^c \mathcal{L}(f(\boldsymbol{x}), j) = C, \quad \forall x \in \mathcal{X}, \forall f. \qquad (3)$$

The main contribution of Ghosh et al. [10] is they proved that if loss function is *symmetric* and $\eta < \frac{c-1}{c}$, then under *uniform* label noise, for any $f$, $R_{\mathcal{L}}^\eta(f^*) - R_{\mathcal{L}}^\eta(f) \leq 0$. Hence, $f^*$ is also the global minimizer for $R_{\mathcal{L}}^\eta$ and $\mathcal{L}$ is noise tolerant. Moreover, if $R_{\mathcal{L}}(f^*) = 0$, then $\mathcal{L}$ is also noise tolerant under class dependent noise.

Being a *nonsymmetric* and *unbounded* loss function, CCE is sensitive to label noise. On the contrary, MAE, as a *symmetric* loss function, is noise robust. For DNNs with a softmax output layer, MAE can be computed as:

$$\mathcal{L}_{MAE}(f(\boldsymbol{x}), \boldsymbol{e}_j) = ||\boldsymbol{e}_j - f(\boldsymbol{x})||_1 = 2 - 2f_j(\boldsymbol{x}). \qquad (4)$$

With this particular configuration of DNN, the proposed MAE loss is, up to a constant of proportionality, the same as the unhinged loss $\mathcal{L}_{unh}(f(\boldsymbol{x}), \boldsymbol{e}_j) = 1 - f_j(\boldsymbol{x})$ [38].

## 3.2 $\mathcal{L}_q$ Loss for Classification

In this section, we will argue that MAE has some drawbacks as a classification loss function for DNNs, which are normally trained on large scale datasets using stochastic gradient based techniques. Let's look at the gradient of the loss functions:

$$\sum_{i=1}^n \frac{\partial \mathcal{L}(f(\boldsymbol{x}_i; \boldsymbol{\theta}), y_i)}{\partial \boldsymbol{\theta}} = \begin{cases} \sum_{i=1}^n -\frac{1}{f_{y_i}(\boldsymbol{x}_i; \boldsymbol{\theta})} \nabla_{\boldsymbol{\theta}} f_{y_i}(\boldsymbol{x}_i; \boldsymbol{\theta}) & \text{for CCE} \\ \sum_{i=1}^n -\nabla_{\boldsymbol{\theta}} f_{y_i}(\boldsymbol{x}_i; \boldsymbol{\theta}) & \text{for MAE/unhinged loss.} \end{cases} \qquad (5)$$

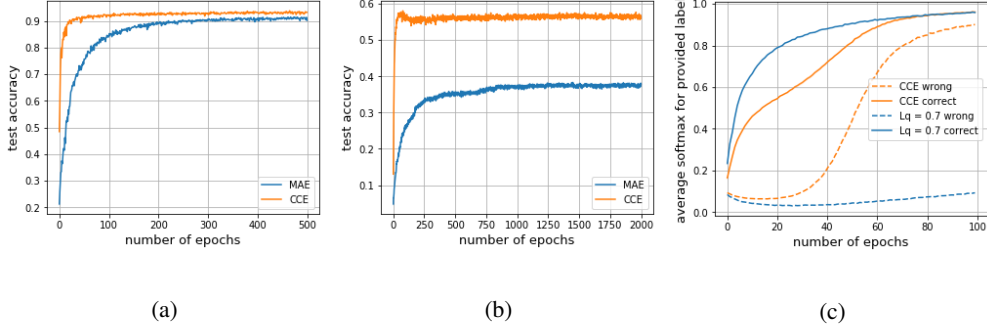

$$\qquad\qquad\text{(a)}\qquad\qquad\qquad\qquad\qquad\text{(b)}\qquad\qquad\qquad\qquad\qquad\text{(c)}$$

Figure 1: (a), (b) Test accuracy against number of epochs for training with CCE (orange) and MAE (blue) loss on clean data with (a) CIFAR-10 and (b) CIFAR-100 datasets. (c) Average softmax prediction for correctly (solid) and wrongly (dashed) labeled training samples, for CCE (orange) and $\mathcal{L}_q$ ($q = 0.7$, blue) loss on CIFAR-10 with uniform noise ($\eta = 0.4$).

Thus, in CCE, samples with softmax outputs that are less congruent with provided labels, and hence smaller $f_{y_i}(\boldsymbol{x}_i; \boldsymbol{\theta})$ or larger $1/f_{y_i}(\boldsymbol{x}_i; \boldsymbol{\theta})$, are implicitly weighed more than samples with predictions that agree more with provided labels in the gradient update. This means that, when training with CCE, more emphasis is put on difficult samples. This implicit weighting scheme is desirable for training with clean data, but can cause overfitting to noisy labels. Conversely, since the $1/f_{y_i}(\boldsymbol{x}_i; \boldsymbol{\theta})$ term is absent in its gradient, MAE treats every sample equally, which makes it more robust to noisy labels. However, as we demonstrate empirically, this can lead to significantly longer training time before convergence. Moreover, without the implicit weighting scheme to focus on challenging samples, the stochasticity involved in the training process can make learning difficult. As a result, classification accuracy might suffer.

To demonstrate this, we conducted a simple experiment using ResNet [13] optimized with the default setting of Adam [19] on the CIFAR datasets [20]. Fig. 1(a) shows the test accuracy curve when trained with CCE and MAE respectively on CIFAR-10. As illustrated clearly, it took significantly longer to converge when trained with MAE. In agreement with our analysis, there was also a compromise in classification accuracy due to the increased difficulty of learning useful features. These adverse effects become much more severe when using a more difficult dataset, such as CIFAR-100 (see Fig. 1(b)). Not only do we observe significantly slower convergence, but also a substantial drop in test accuracy when using MAE. In fact, the maximum test accuracy achieved after 2000 epochs, a long time after training using CCE has converged, was 38.29%, while CCE achieved an higher accuracy of 39.92% after merely 7 epochs! Despite its theoretical noise-robustness, due to the shortcoming during training induced by its noise-robustness, we conclude that MAE is not suitable for DNNs with challenging datasets like ImageNet.

To exploit the benefits of both the noise-robustness provided by MAE and the implicit weighting scheme of CCE, we propose using the the negative Box-Cox transformation [4] as a loss function:

$$\mathcal{L}_q(f(\boldsymbol{x}), \boldsymbol{e}_j) = \frac{(1 - f_j(\boldsymbol{x})^q)}{q}, \qquad (6)$$

where $q \in (0, 1]$. Using L'Hôpital's rule, it can be shown that the proposed loss function is equivalent to CCE for $\lim_{q \to 0} \mathcal{L}_q(f(\boldsymbol{x}), \boldsymbol{e}_j)$, and becomes MAE/unhinged loss when $q = 1$. Hence, this loss is a generalization of CCE and MAE. Relatedly, Ferrari and Yang [7] viewed the maximization of Eq. 6 as a generalization of maximum likelihood and termed the loss function $\mathcal{L}_q$, which we also adopt.

Theoretically, for any input $\boldsymbol{x}$, the sum of $\mathcal{L}_q$ loss with respect to all classes is bounded by:

$$\frac{c - c^{(1-q)}}{q} \leq \sum_{j=1}^{c} \frac{(1 - f_j(\boldsymbol{x})^q)}{q} \leq \frac{c - 1}{q}. \qquad (7)$$

Using this bound and under uniform noise with $\eta \leq 1 - \frac{1}{c}$, we can show (see Appendix)

$$A \leq (R_{\mathcal{L}_q}(f^*) - R_{\mathcal{L}_q}(\hat{f})) \leq 0, \qquad (8)$$

where $A = \frac{\eta[1-c^{(1-q)}]}{q(c-1-\eta c)} < 0$, $f^*$ is the global minimizer of $R_{\mathcal{L}_q}(f)$, and $\hat{f}$ is the global minimizer of $R^\eta_{\mathcal{L}_q}(f)$. The larger the $q$, the larger the constant $A$, and the tighter the bound of Eq. 8. In the extreme case of $q = 1$ (i.e., for MAE), $A = 0$ and $R_{\mathcal{L}_q}(\hat{f}) = R_{\mathcal{L}_q}(f^*)$. In other words, for $q$ values approaching 1, the optimum of the noisy risk will yield a risk value (on the clean data) that is close to $f^*$, which implies noise tolerance. It can also be shown that the difference $(R^\eta_{\mathcal{L}_q}(f^*) - R^\eta_{\mathcal{L}_q}(\hat{f}))$ is bounded under class dependent noise, provided $R_{\mathcal{L}_q}(f^*) = 0$ and $q_{ij} < q_{ii} \ \forall i \neq j$ (see Thm 2 in Appendix).

The compromise on noise-robustness when using $\mathcal{L}_q$ over MAE prompts an easier learning process. Let's look at the gradients of $\mathcal{L}_q$ loss to see this:

$$\frac{\partial \mathcal{L}_q(f(\boldsymbol{x}_i; \boldsymbol{\theta}), y_i)}{\partial \boldsymbol{\theta}} = f_{y_i}(\boldsymbol{x}_i; \boldsymbol{\theta})^q (-\frac{1}{f_{y_i}(\boldsymbol{x}_i; \boldsymbol{\theta})} \nabla_{\boldsymbol{\theta}} f_{y_i}(\boldsymbol{x}_i; \boldsymbol{\theta})) = -f_{y_i}(\boldsymbol{x}_i; \boldsymbol{\theta})^{q-1} \nabla_{\boldsymbol{\theta}} f_{y_i}(\boldsymbol{x}_i; \boldsymbol{\theta}),$$

where $f_{y_i}(\boldsymbol{x}_i; \boldsymbol{\theta}) \in [0, 1] \ \forall i$ and $q \in (0, 1)$. Thus, relative to CCE, $\mathcal{L}_q$ loss weighs each sample by an additional $f_{y_i}(\boldsymbol{x}_i; \boldsymbol{\theta})^q$ so that less emphasis is put on samples with weak agreement between softmax outputs and the labels, which should improve robustness against noise. Relative to MAE, a weighting of $f_{y_i}(\boldsymbol{x}_i; \boldsymbol{\theta})^{q-1}$ on each sample can facilitate learning by giving more attention to challenging datapoints with labels that do not agree with the softmax outputs. On one hand, larger $q$ leads to a more noise-robust loss function. On the other hand, too large of a $q$ can make optimization strenuous. Hence, as we will demonstrate empirically below, it is practically useful to set $q$ between 0 and 1, where a tradeoff equilibrium is achieved between noise-robustness and better learning dynamics.

### 3.3 Truncated $\mathcal{L}_q$ Loss

Since a tighter bound in $\sum_{j=1}^c \mathcal{L}(f(\boldsymbol{x}, j))$ would imply stronger noise tolerance, we propose the truncated $\mathcal{L}_q$ loss:

$$\mathcal{L}_{trunc}(f(\boldsymbol{x}), \boldsymbol{e}_j) = \begin{cases} \mathcal{L}_q(k) & \text{if } f_j(\boldsymbol{x}) \leq k \\ \mathcal{L}_q(f(\boldsymbol{x}), \boldsymbol{e}_j) & \text{if } f_j(\boldsymbol{x}) > k \end{cases} \tag{9}$$

where $0 < k < 1$, and $\mathcal{L}_q(k) = (1 - k^q)/q$. Note that, when $k \to 0$, the truncated $\mathcal{L}_q$ loss becomes the normal $\mathcal{L}_q$ loss. Assuming $k \geq 1/c$, the sum of truncated $\mathcal{L}_q$ loss with respect to all classes is bounded by (see Appendix):

$$d\mathcal{L}_q(\frac{1}{d}) + (c - d)\mathcal{L}_q(k) \leq \sum_{j=1}^c \mathcal{L}_{trunc}(f(\boldsymbol{x}), \boldsymbol{e}_j) \leq c\mathcal{L}_q(k), \tag{10}$$

where $d = \max(1, \frac{(1-q)^{1/q}}{k})$. It can be verified that the difference between upper and lower bounds for the truncated $\mathcal{L}_q$ loss, $\mathcal{L}_q(k)$, is smaller than that for the $\mathcal{L}_q$ loss of Eq. 7, if

$$d[\mathcal{L}_q(k) - \mathcal{L}_q(\frac{1}{d})] < \frac{c^{(1-q)} - 1}{q}. \tag{11}$$

As an example, when $k \geq 0.3$, the above inequality is satisfied for all $q$ and $c$. When $k \geq 0.2$, the inequality is satisfied for all q and $c \geq 10$. Since the derived bounds in Eq. 7 and Eq. 10 are tight, introducing the threshold $k$ can thus lead to a more noise tolerant loss function.

If the softmax output for the provided label is below a threshold, truncated $\mathcal{L}_q$ loss becomes a constant. Thus, the loss gradient is zero for that sample, and it does not contribute to learning dynamics. While Eq. 10 suggests that a larger threshold $k$ leads to tighter bounds and hence more noise-robustness, too large of a threshold would precipitate too many discarded samples for training. Ideally, we would want the algorithm to train with all available clean data and ignore noisy labels. Thus the optimal choice of $k$ would depend on the noise in the labels. Hence, $k$ can be treated as a (bounded) hyper-parameter and optimized. In our experiments, we set $k = 0.5$ that yields a tighter bound for truncated $\mathcal{L}_q$ loss, and which we observed to work well empirically.

A potential problem arises when training directly with this loss function. When the threshold is relatively large (e.g., $k = 0.5$ in a 10-class classification problem), at the beginning of the training phase, most of the softmax outputs can be significantly smaller than $k$, resulting in a dramatic drop

in the number of effective samples. Moreover, it is suboptimal to prune samples based on softmax values at the beginning of training. To circumvent the problem, observe that, by definition of the truncated $\mathcal{L}_q$ loss:

$$\operatorname*{argmin}_{\boldsymbol{\theta}} \sum_{i=1}^{n} \mathcal{L}_{trunc}(f(\boldsymbol{x}_i; \boldsymbol{\theta}), y_i) = \operatorname*{argmin}_{\boldsymbol{\theta}} \sum_{i=1}^{n} v_i \mathcal{L}_q(f(\boldsymbol{x}_i; \boldsymbol{\theta}), y_i) + (1 - v_i)\mathcal{L}_q(k), \qquad (12)$$

where $v_i = 0$ if $f_{y_i}(\mathbf{x}_i) \leq k$ and $v_i = 1$ otherwise, and $\boldsymbol{\theta}$ represents the parameters of the classifier. Optimizing the above loss is the same as optimizing the following:

$$\operatorname*{argmin}_{\boldsymbol{\theta}} \sum_{i=1}^{n} v_i \mathcal{L}_q(f(\boldsymbol{x}_i; \boldsymbol{\theta}), y_i) - v_i \mathcal{L}_q(k) = \operatorname*{argmin}_{\boldsymbol{\theta}, \boldsymbol{w} \in [0,1]^n} \sum_{i=1}^{n} w_i \mathcal{L}_q(f(\boldsymbol{x}_i; \boldsymbol{\theta}), y_i) - \mathcal{L}_q(k) \sum_{i=1}^{n} w_i,$$
$$(13)$$

because for any $\boldsymbol{\theta}$, the optimal $w_i$ is 1 if $\mathcal{L}_q(f(\boldsymbol{x}_i; \boldsymbol{\theta}), y_i) \leq \mathcal{L}_q(k)$ and 0 if $\mathcal{L}_q(f(\boldsymbol{x}_i; \boldsymbol{\theta}), y_i) > \mathcal{L}_q(k)$. Hence, we can optimize the truncated $\mathcal{L}_q$ loss by optimizing the right hand side of Eq. 13. If $\mathcal{L}_q$ is convex with respect to the parameters $\boldsymbol{\theta}$, optimizing Eq. 13 is a *biconvex optimization* problem, and the alternative convex search (ACS) algorithm [3] can be used to find the global minimum. ACS iteratively optimizes $\boldsymbol{\theta}$ and $\boldsymbol{w}$ while keeping the other set of parameters fixed. Despite the high non-convexity of DNNs, we can apply ACS to find a local minimum. We refer to the update of $\boldsymbol{w}$ as "pruning". At every step of iteration, pruning can be carried out easily by computing $f(\boldsymbol{x}_i; \boldsymbol{\theta}^{(t)})$ for all training samples. Only samples with $f_{y_i}(\boldsymbol{x}_i; \boldsymbol{\theta}^{(t)}) \geq k$ and $\mathcal{L}_q(f(\boldsymbol{x}_i; \boldsymbol{\theta}), y_i) \leq \mathcal{L}_q(k)$ are kept for updating $\boldsymbol{\theta}$ during that iteration (and hence $w_i = 1$ ). The additional computational complexity from the pruning steps is negligible. Interestingly, the resulting algorithm is similar to that of self-paced learning [22].

---

**Algorithm 1** ACS for Training with $\mathcal{L}_q$ Loss

---

**Input** Noisy dataset $D_\eta$, total iterations $T$, threshold $k$

    Initialize $w_i^{(0)} = 1 \ \forall \ i$
    Update $\boldsymbol{\theta}^{(0)} = \operatorname{argmin}_{\boldsymbol{\theta}} \sum_{i=1}^{n} w_i^{(0)} \mathcal{L}_q(f(\boldsymbol{x}_i; \boldsymbol{\theta}), y_i) - \mathcal{L}_q(k) \sum_{i=1}^{n} w_i^{(0)}$
    **while** $t < T$ **do**
        Update $\boldsymbol{w}^{(t)} = \operatorname{argmin}_{\boldsymbol{w}} \sum_{i=1}^{n} w_i \mathcal{L}_q(f(\boldsymbol{x}_i; \boldsymbol{\theta}^{(t-1)}), y_i) - \mathcal{L}_q(k) \sum_{i=1}^{n} w_i$ [Pruning Step]
        Update $\boldsymbol{\theta}^{(t)} = \operatorname{argmin}_{\boldsymbol{\theta}} \sum_{i=1}^{n} w_i^{(t)} \mathcal{L}_q(f(\boldsymbol{x}_i; \boldsymbol{\theta}), y_i) - \mathcal{L}_q(k) \sum_{i=1}^{n} w_i^{(t)}$
**Output** $\boldsymbol{\theta}^{(T)}$

---

## 4 Experiments

The following setup applies to all of the experiments conducted. Noisy datasets were produced by artificially corrupting true labels. 10% of the training data was retained for validation. To realistically mimic a noisy dataset while justifiably analyzing the performance of the proposed loss function, only the training and validation data were contaminated, and test accuracies were computed with respect to true labels. A mini-batch size of 128 was used. All networks used ReLUs in the hidden layers and softmax layers at the output. All reported experiments were repeated five times with random initialization of neural network parameters and randomly generated noisy labels each time. We compared the proposed functions with CCE, MAE and also the confusion matrix-corrected CCE, as shown in Eq. 1. Following [32], we term this "forward correction". All experiments were conducted with identical optimization procedures and architectures, changing only the loss functions.

### 4.1 Toward a Better Understanding of $\mathcal{L}_q$ Loss

To better grasp the behavior of $\mathcal{L}_q$ loss, we implemented different values of $q$ and uniform noise at different noise levels, and trained ResNet-34 with the default setting of Adam on CIFAR-10. As shown in Fig. 2, when trained on clean dataset, increasing $q$ not only slowed down the rate of convergence, but also lowered the classification accuracy. More interesting phenomena appeared when trained on noisy data. When CCE ($q = 0$) was used, the classifier first learned predictive

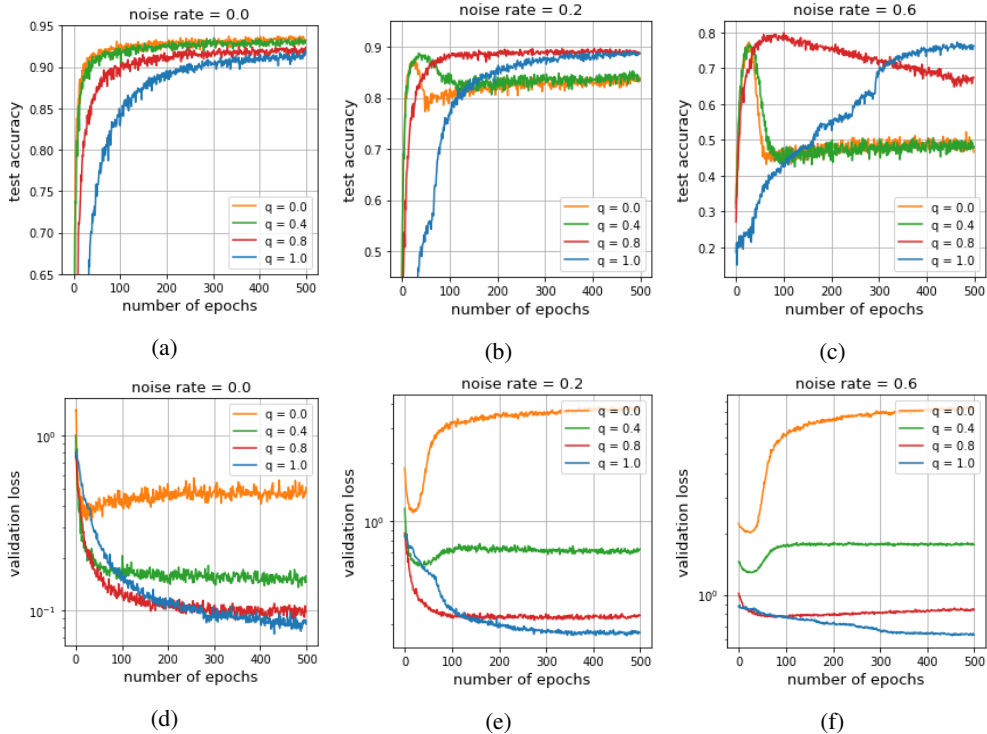

Figure 2: The test accuracy and validation loss against number of epochs for training with $\mathcal{L}_q$ loss at different values of $q$. (a) and (d): $\eta = 0.0$; (b) and (e): $\eta = 0.2$; (c) and (f): $\eta = 0.6$.

patterns, presumably from the noise-free labels, before overfitting strongly to the noisy labels, in agreement with Arpit et al.'s observations [1]. Training with increased $q$ values delayed overfitting and attained higher classification accuracies. One interpretation of this behavior is that the classifier could learn more about predictive features before overfitting. This interpretation is supported by our plot of the average softmax values with respect to the correctly and wrongly labeled samples on the training set for CCE and $\mathcal{L}_q$ ($q = 0.7$) loss, and with $40\%$ uniform noise (Fig. 1(c)). For CCE, the average softmax for wrongly labeled samples remained small at the beginning, but grew quickly when the model started overfitting. $\mathcal{L}_q$ loss, on the other hand, resulted in significantly smaller softmax values for wrongly labeled data. This observation further serves as an empirical justification for the use of truncated $\mathcal{L}_q$ loss as described in section 3.3.

We also observed that there was a threshold of $q$ beyond which overfitting never kicked in before convergence. When $\eta = 0.2$ for instance, training with $\mathcal{L}_q$ loss with $q = 0.8$ produced an overfitting-free training process. Empirically, we noted that, the noisier the data, the larger this threshold is. However, too large of a $q$ hampers the classification accuracy, and thus a larger $q$ is not always preferred. In general, $q$ can be treated as a hyper-parameter that can be optimized, say via monitoring validation accuracy. In remaining experiments, we used $q = 0.7$, which yielded a good compromise between fast convergence and noise robustness (no overfitting was observed for $\eta \leq 0.5$).

## 4.2 Datasets

**CIFAR-10/CIFAR-100**: ResNet-34 was used as the classifier optimized with the loss functions mentioned above. Per-pixel mean subtraction, horizontal random flip and $32 \times 32$ random crops after padding with 4 pixels on each side was performed as data preprocessing and augmentation. Following [15], we used stochastic gradient descent (SGD) with $0.9$ momentum, a weight decay of $10^{-4}$ and learning rate of $0.01$, and divided it by 10 after 40 and 80 epochs (120 in total) for CIFAR-10, and after 80 and 120 (150 in total) for CIFAR-100. To ensure a fair comparison, the identical optimization scheme was used for truncated $\mathcal{L}_q$ loss. We trained with the entire dataset for the first 40 epochs for CIFAR-10 and 80 for CIFAR-100, and started pruning and training with the pruned dataset afterwards. Pruning was done every 10 epochs. To prevent overfitting, we used the model at the optimal epoch

Table 1: Average test accuracy and standard deviation (5 runs) on experiments with closed-set noise. We report accuracies of the epoch where validation accuracy is maximum. Forward $T$ and $\hat{T}$ represent forward correction with the true and estimated confusion matrices, respectively [32]. $q = 0.7$ was used for all experiments with $\mathcal{L}_q$ loss and truncated $\mathcal{L}_q$ loss. Best 2 accuracies are **bold faced**.

| Datasets | Loss Functions | Uniform Noise | | | | Class Dependent Noise | | | |
|---|---|---|---|---|---|---|---|---|---|
| | | Noise Rate $\eta$ | | | | Noise Rate $\eta$ | | | |
| | | 0.2 | 0.4 | 0.6 | 0.8 | 0.1 | 0.2 | 0.3 | 0.4 |
| FASHION MNIST | CCE | 93.24 ± 0.12 | 92.09 ± 0.18 | 90.29 ± 0.35 | 86.20 ± 0.68 | 94.06 ± 0.05 | 93.72 ± 0.14 | 92.72 ± 0.21 | 89.82 ± 0.31 |
| | MAE | 80.39 ± 4.68 | 79.30 ± 6.20 | 82.41 ± 5.29 | 74.73 ± 5.26 | 74.03 ± 6.32 | 63.03 ± 3.91 | 58.14 ± 0.14 | 56.04 ± 3.76 |
| | Forward $T$ | **93.64 ± 0.12** | **92.69 ± 0.20** | **91.16 ± 0.16** | 87.59 ± 0.35 | **94.33 ± 0.10** | **94.03 ± 0.11** | **93.91 ± 0.14** | **93.65 ± 0.11** |
| | Forward $\hat{T}$ | 93.26 ± 0.10 | 92.24 ± 0.15 | 90.54 ± 0.10 | 85.57 ± 0.86 | **94.09 ± 0.10** | **93.66 ± 0.09** | **93.52 ± 0.16** | 88.53 ± 4.81 |
| | $\mathcal{L}_q$ | **93.35 ± 0.09** | 92.58 ± 0.11 | 91.30 ± 0.20 | **88.01 ± 0.22** | 93.51 ± 0.17 | 93.24 ± 0.14 | 92.21 ± 0.27 | 89.53 ± 0.53 |
| | Trunc $\mathcal{L}_q$ | 93.21 ± 0.05 | **92.60 ± 0.17** | **91.56 ± 0.16** | **88.33 ± 0.38** | 93.53 ± 0.11 | 93.36 ± 0.07 | 92.76 ± 0.14 | **91.62 ± 0.34** |
| CIFAR-10 | CCE | 86.98 ± 0.44 | 81.88 ± 0.29 | 74.14 ± 0.56 | 53.82 ± 1.04 | 90.69 ± 0.17 | 88.59 ± 0.34 | 86.14 ± 0.40 | 80.11 ± 1.44 |
| | MAE | 83.72 ± 3.84 | 67.00 ± 4.45 | 64.21 ± 5.28 | 38.63 ± 2.62 | 82.61 ± 4.81 | 52.93 ± 3.60 | 50.36 ± 5.55 | 45.52 ± 0.13 |
| | Forward $T$ | 88.63 ± 0.14 | 85.07 ± 0.29 | 79.12 ± 0.64 | **64.30 ± 0.70** | **91.32 ± 0.21** | **90.35 ± 0.26** | **89.25 ± 0.43** | **88.12 ± 0.32** |
| | Forward $\hat{T}$ | 87.99 ± 0.36 | 83.25 ± 0.38 | 74.96 ± 0.65 | 54.64 ± 0.44 | 90.52 ± 0.26 | 89.09 ± 0.47 | 86.79 ± 0.36 | **83.55 ± 0.58** |
| | $\mathcal{L}_q$ | **89.83 ± 0.20** | **87.13 ± 0.22** | 82.54 ± 0.23 | 64.07 ± 1.38 | **90.91 ± 0.22** | 89.33 ± 0.17 | 85.45 ± 0.74 | 76.74 ± 0.61 |
| | Trunc $\mathcal{L}_q$ | **89.7 ± 0.11** | **87.62 ± 0.26** | **82.70 ± 0.23** | **67.92 ± 0.60** | 90.43 ± 0.25 | **89.45 ± 0.29** | **87.10 ± 0.22** | 82.28 ± 0.67 |
| CIFAR-100 | CCE | 58.72 ± 0.26 | 48.20 ± 0.65 | 37.41 ± 0.94 | 18.10 ± 0.82 | 66.54 ± 0.42 | 59.20 ± 0.18 | 51.40 ± 0.16 | 42.74 ± 0.61 |
| | MAE | 15.80 ± 1.38 | 9.03 ± 1.54 | 7.74 ± 1.48 | 3.76 ± 0.27 | 13.38 ± 1.84 | 11.50 ± 1.16 | 8.91 ± 0.89 | 8.20 ± 1.04 |
| | Forward $T$ | 63.16 ± 0.37 | 54.65 ± 0.88 | 44.62 ± 0.82 | 24.83 ± 0.71 | **71.05 ± 0.30** | **71.08 ± 0.22** | **70.76 ± 0.26** | **70.82 ± 0.45** |
| | Forward $\hat{T}$ | 39.19 ± 2.61 | 31.05 ± 1.44 | 19.12 ± 1.95 | 8.99 ± 0.58 | 45.96 ± 1.21 | 42.46 ± 2.16 | 38.13 ± 2.97 | 34.44 ± 1.93 |
| | $\mathcal{L}_q$ | **66.81 ± 0.42** | **61.77 ± 0.24** | **53.16 ± 0.78** | **29.16 ± 0.74** | 68.36 ± 0.42 | **66.59 ± 0.22** | **61.45 ± 0.26** | **47.22 ± 1.15** |
| | Trunc $\mathcal{L}_q$ | **67.61 ± 0.18** | **62.64 ± 0.33** | **54.04 ± 0.56** | **29.60 ± 0.51** | **68.86 ± 0.14** | **66.59 ± 0.23** | **61.87 ± 0.39** | **47.66 ± 0.69** |

Table 2: Average test accuracy on experiments with CIFAR-10. We replicated the exact experimental setup as in [40]. The reported accuracies are the average last epoch accuracies after training for 100 epochs. $\eta = 40\%$. CCE, Forward and method by Wang et al. are adapted for direct comparison.

| Noise type | CCE [40] | Forward [40] | Wang, et al. [40] | MAE | $\mathcal{L}_q$ | Trunc $\mathcal{L}_q$ |
|---|---|---|---|---|---|---|
| CIFAR-10 + CIFAR-100 (open-set noise) | 62.92 | 64.18 | 79.28 | 75.06 | 71.10 | **79.55** |
| CIFAR-10 (closed-set noise) | 62.38 | 77.81 | 78.15 | 74.31 | 64.79 | **79.12** |

based on maximum validation accuracy for pruning. Uniform noise was generated by mapping a true label to a random label through uniform sampling. Following Patrini, et al. [32] class dependent noise was generated by mapping TRUCK → AUTOMOBILE, BIRD → AIRPLANE, DEER → HORSE, and CAT ↔ DOG with probability $\eta$ for CIFAR-10. For CIFAR-100, we simulated class-dependent noise by flipping each class into the next circularly with probability $\eta$.

We also tested noise-robustness of our loss function on open-set noise using CIFAR-10. For a direct comparison, we followed the identical setup as described in [40]. For this experiment, the classifier was trained for only 100 epochs. We observed validation loss plateaued after about 10 epochs, and hence started pruning the data afterwards at 10-epoch intervals. The open-set noise was generated by using images from the CIFAR-100 dataset. A random CIFAR-10 label was assigned to these images.

**FASHION-MNIST**: ResNet-18 was used. The identical data preprocessing, augmentation, and optimization procedure as in CIFAR-10 was deployed for training. To generate a realistic class dependent noise, we used the t-SNE [25] plot of the dataset to associated classes with similar embeddings, and mapped BOOT → SNEAKER , SNEAKER → SANDALS, PULLOVER → SHIRT, COAT ↔ DRESS with probability $\eta$.

## 4.3 Results and Discussion

Experimental results with closed-set noise is summarized in Table 1. For uniform noise, proposed loss functions outperformed the baselines significantly, including forward correction with the ground truth confusion matrices. In agreement with our theoretical expectations, truncating the $\mathcal{L}_q$ loss enhanced results. For class dependent noise, in general Forward $T$ offered the best performance, as it relied on the knowledge of the ground truth confusion matrix. Truncated $\mathcal{L}_q$ loss produced similar accuracies as Forward $\hat{T}$ for FASHION-MNIST and better results for CIFAR datasets, and outperformed the other baselines at most noise levels for all datasets. While using $\mathcal{L}_q$ loss improved over baselines for CIFAR-100, no improvements were observed for FASHION-MNIST and CIFAR-10 datasets. We believe this is in part because very similar classes were grouped together for the confusion matrices and consquently the DNNs might falsely put high confidence on wrongly labeled samples.

In general, classification accuracy for both uniform and class dependent noise would be further improved relative to baselines with optimized $q$ and $k$ values and more number of epochs. Based on the experimental results, we believe the proposed approach would work well when correctly labeled data can be differentiated from wrongly labeled data based on softmax outputs, which is often the case with large-scale data and expressive models. We also observed that MAE performed poorly for all datasets at all noise levels, presumably because DNNs like ResNet struggled to optimize with MAE loss, especially on challenging datasets such as CIFAR-100.

Table 2 summarizes the results for open-set noise with CIFAR-10. Following Wang et al. [40], we reported the last-epoch test accuracy after training for 100 epochs. We also repeated the closed-set noise experiment with their setup. Using $\mathcal{L}_q$ loss noticeably prevented overfitting, and using truncated $\mathcal{L}_q$ loss achieved better results than the state-of-the-art method for open-set noise reported in [40]. Moreover, our method is significantly easier to implement. Lastly, note that the poor performance of $\mathcal{L}_q$ loss compared to MAE is due to the fact that test accuracy reported here is long after the model started overfitting, since a shallow CNN without data augmentation was deployed for this experiment.

## 5    Conclusion

In conclusion, we proposed theoretically grounded and easy-to-use classes of noise-robust loss functions, the $\mathcal{L}_q$ loss and the truncated $\mathcal{L}_q$ loss, for classification with noisy labels that can be employed with any existing DNN algorithm. We empirically verified noise robustness on various datasets with both closed- and open-set noise scenarios.

### Acknowledgments

This work was supported by NIH R01 grants (R01LM012719 and R01AG053949), the NSF NeuroNex grant 1707312, and NSF CAREER grant (1748377).

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
