[Supplementary Material]

# Appendix

**Lemma 1.** $\lim_{q \to 0} \mathcal{L}_q(f(\boldsymbol{x}), \boldsymbol{e}_j) = \mathcal{L}_C(f(\boldsymbol{x}), \boldsymbol{e}_j)$, where $\mathcal{L}_q$ represents the $\mathcal{L}_q$ loss, and $\mathcal{L}_C$ represents the categorical cross entropy loss.

*Proof.* from equation 6, and using L'Hôpital's rule,

$$
\lim_{q \to 0} \mathcal{L}_q(f(\boldsymbol{x}), \boldsymbol{e}_j) = \lim_{q \to 0} \frac{(1 - f_j(\boldsymbol{x})^q)}{q} = \lim_{q \to 0} \frac{\frac{d}{dq}(1 - f_j(\boldsymbol{x})^q)}{\frac{d}{dq}q}
$$
$$
= \lim_{q \to 0} -f_j(\boldsymbol{x})^q \log(f_j(\boldsymbol{x})) = -\log(f_j(\boldsymbol{x})) = \mathcal{L}_C(f(\boldsymbol{x}), \boldsymbol{e}_j).
$$

$\square$

**Lemma 2.** *For any $\boldsymbol{x}$ and $q \in (0, 1]$, the sum of $\mathcal{L}_q$ loss with respect to all classes is bounded by:*

$$
\frac{c - c^{(1-q)}}{q} \le \sum_{j=1}^{c} \frac{(1 - f_j(\boldsymbol{x})^q)}{q} \le \frac{c - 1}{q}. \tag{14}
$$

*Proof.* Observe that, since we have a softmax layer at the end, $f_j(\boldsymbol{x}) \le 1$ for all $j$, and $\sum_{j=1}^{c} f_j(\boldsymbol{x}) = 1$. Now, since $q \in (0, 1]$, we have $f_j(\boldsymbol{x}) \le f_j(\boldsymbol{x})^q$, and $(1 - f_j(\boldsymbol{x})) \ge (1 - f_j(\boldsymbol{x})^q)$. Hence,

$$
\sum_{j=1}^{c} \frac{(1 - f_j(\boldsymbol{x})^q)}{q} \le \sum_{j=1}^{c} \frac{(1 - f_j(\boldsymbol{x}))}{q} = \frac{c - \sum_{j=1}^{c} f_j(\boldsymbol{x})}{q} = \frac{c - 1}{q}.
$$

Moreover, since $\sum_{j=1}^{c} f_j(\boldsymbol{x})^q \le \sum_{j=1}^{c}(1/c)^q$ for all $\boldsymbol{x}$ and $q \in (0, 1]$, $\sum_{j=1}^{c}(1 - f_j(\boldsymbol{x})^q) \ge \sum_{j=1}^{c}(1 - (1/c)^q)$, and

$$
\sum_{j=1}^{c} \frac{(1 - f_j(\boldsymbol{x})^q)}{q} \ge \sum_{j=1}^{c} \frac{(1 - (1/c)^q)}{q} = \frac{c - c^{(1-q)}}{q}.
$$

$\square$

**Theorem 1.** *Under uniform noise with $\eta \le 1 - \frac{1}{c}$,*

$$
0 \le (R_{\mathcal{L}_q}^{\eta}(f^*) - R_{\mathcal{L}_q}^{\eta}(\hat{f})) \le A, \tag{15}
$$

*and*

$$
A' \le R_{\mathcal{L}_q}(f^*) - R_{\mathcal{L}_q}(\hat{f}) \le 0, \tag{16}
$$

*where $A = \frac{\eta[c^{(1-q)}-1]}{q(c-1)} \ge 0$, $A' = \frac{\eta[1-c^{(1-q)}]}{q(c-1-\eta c)} < 0$, $f^*$ is the global minimizer of $R_{\mathcal{L}_q}(f)$, and $\hat{f}$ is the global minimizer of $R_{\mathcal{L}_q}^{\eta}(f)$.*

*Proof.* Recall that for any softmax output $f$,

$$
R_{\mathcal{L}_q}(f) = \mathbb{E}_D[\mathcal{L}_q(f(\boldsymbol{x}), y_{\boldsymbol{x}})] = \mathbb{E}_{\boldsymbol{x}, y_{\boldsymbol{x}}}[\mathcal{L}_q(f(\boldsymbol{x}), y_{\boldsymbol{x}})],
$$

and since for uniform noise with noise rate $\eta$, $\eta_{jk} = 1 - \eta$ for $j = k$, and $\eta_{jk} = \frac{\eta}{c-1}$ for $j \ne k$, we have

$$R^{\eta}_{\mathcal{L}_q}(f) = \mathbb{E}_D[\mathcal{L}_q(f(\boldsymbol{x}), \widetilde{y}_{\boldsymbol{x}})] = \mathbb{E}_{\boldsymbol{x}, \widetilde{y}_{\boldsymbol{x}}}[\mathcal{L}_q(f(\boldsymbol{x}), \widetilde{y}_{\boldsymbol{x}})]$$

$$= \mathbb{E}_{\boldsymbol{x}} \mathbb{E}_{y_{\boldsymbol{x}}|\boldsymbol{x}} \mathbb{E}_{\widetilde{y}_{\boldsymbol{x}}|y_{\boldsymbol{x}}, \boldsymbol{x}}[\mathcal{L}_q(f(\boldsymbol{x}), \widetilde{y}_{\boldsymbol{x}})]$$

$$= \mathbb{E}_{\boldsymbol{x}} \mathbb{E}_{y_{\boldsymbol{x}}|\boldsymbol{x}}[(1-\eta)\mathcal{L}_q(f(\boldsymbol{x}), y_{\boldsymbol{x}}) + \frac{\eta}{c-1} \sum_{i \neq y_{\boldsymbol{x}}} \mathcal{L}_q(f(\boldsymbol{x}), i)]$$

$$= \mathbb{E}_{\boldsymbol{x}} \mathbb{E}_{y_{\boldsymbol{x}}|\boldsymbol{x}}[(1-\eta)\mathcal{L}_q(f(\boldsymbol{x}), y_{\boldsymbol{x}}) + \frac{\eta}{c-1}(\sum_{i=1}^{c} \mathcal{L}_q(f(\boldsymbol{x}), i) - \mathcal{L}_q(f(\boldsymbol{x}), y_{\boldsymbol{x}}))]$$

$$= (1-\eta)R_{\mathcal{L}_q}(f) - \frac{\eta}{c-1}R_{\mathcal{L}_q}(f) + \frac{\eta}{c-1}\mathbb{E}_{\boldsymbol{x}} \mathbb{E}_{y_{\boldsymbol{x}}|\boldsymbol{x}}[\sum_{i=1}^{c} \mathcal{L}_q(f(\boldsymbol{x}), i)]$$

$$= (1 - \frac{\eta c}{c-1})R_{\mathcal{L}_q}(f) + \frac{\eta}{c-1}\mathbb{E}_{\boldsymbol{x}} \mathbb{E}_{y_{\boldsymbol{x}}|\boldsymbol{x}}[\sum_{i=1}^{c} \mathcal{L}_q(f(\boldsymbol{x}), i)]$$

Now, from Lemma 2, we have:

$$(1 - \frac{\eta c}{c-1})R_{\mathcal{L}_q}(f) + \frac{\eta[c - c^{(1-q)}]}{q(c-1)} \leq R^{\eta}_{\mathcal{L}_q}(f) \leq (1 - \frac{\eta c}{c-1})R_{\mathcal{L}_q}(f) + \frac{\eta}{q}.$$

We can also write the inequality in terms of $R_{\mathcal{L}_q}(f)$:

$$(R^{\eta}_{\mathcal{L}_q}(f) - \frac{\eta}{q})/(1 - \frac{\eta c}{c-1}) \leq R_{\mathcal{L}_q}(f)) \leq (R^{\eta}_{\mathcal{L}_q}(f) - \frac{\eta[c - c^{(1-q)}]}{q(c-1)})/(1 - \frac{\eta c}{c-1})$$

Thus, for $\hat{f}$,

$$R^{\eta}_{\mathcal{L}_q}(f^*) - R^{\eta}_{\mathcal{L}_q}(\hat{f}) \leq A + (1 - \frac{\eta c}{c-1})(R_{\mathcal{L}_q}(f^*) - R_{\mathcal{L}_q}(\hat{f})) \leq A,$$

or equivalently,

$$R_{\mathcal{L}_q}(f^*) - R_{\mathcal{L}_q}(\hat{f}) \geq A' + (R^{\eta}_{\mathcal{L}_q}(f^*) - R^{\eta}_{\mathcal{L}_q}(\hat{f}))/(1 - \frac{\eta c}{c-1}) \geq A'$$

where $A = \frac{\eta[c^{(1-q)}-1]}{q(c-1)} \geq 0$ and $A' = \frac{\eta[1-c^{(1-q)}]}{q(c-1-\eta c)}$, since $\eta \leq \frac{c-1}{c}$, and $f^*$ is a minimizer of $R_{\mathcal{L}_q}(f)$. Lastly, since $\hat{f}$ is the minimizer of $R^{\eta}_{\mathcal{L}_q}(f)$, we have that $R^{\eta}_{\mathcal{L}_q}(f^*) - R^{\eta}_{\mathcal{L}_q}(\hat{f}) \geq 0$, or $R_{\mathcal{L}_q}(f^*) - R_{\mathcal{L}_q}(\hat{f}) \leq 0$. This completes the proof. $\qquad \square$

**Remark.** *Note that, when $q = 1$, $A = 0$, and $f^*$ is also minimizer of risk under uniform noise.*

**Theorem 2.** *Under class dependent noise when $\eta_{ij} < (1 - \eta_i)$, $\forall j \neq i$, $\forall i, j \in [c]$, where $\eta_{ij} = p(\widetilde{y} = j|y = i)$, $\forall j \neq i$, and $(1 - \eta_i) = p(\widetilde{y} = i|y = i)$, if $R_{\mathcal{L}_q}(f^*) = 0$, then*

$$0 \leq (R^{\eta}_{\mathcal{L}_q}(f^*) - R^{\eta}_{\mathcal{L}_q}(\hat{f})) \leq B, \tag{17}$$

*where $B = \frac{c^{1-q}-1}{q}\mathbb{E}_D(1 - \eta_{y_{\boldsymbol{x}}}) \geq 0$, $f^*$ is the global minimizer of $R_{\mathcal{L}_q}(f)$, and $\hat{f}$ is the global minimizer of $R^{\eta}_{\mathcal{L}_q}(f)$.*

*Proof.* For class dependent noise, from Lemma 2, for any soft-max output function $f$ we have

$$R^{\eta}_{\mathcal{L}_q}(f) = \mathbb{E}_D[(1 - \eta_{y_{\boldsymbol{x}}})\mathcal{L}_q(f(\boldsymbol{x}), y_{\boldsymbol{x}})] + \mathbb{E}_D[\sum_{i \neq y_{\boldsymbol{x}}} \eta_{y_{\boldsymbol{x}} i}\mathcal{L}_q(f(\boldsymbol{x}), i)]$$

$$\leq \mathbb{E}_D[(1 - \eta_{y_{\boldsymbol{x}}})(\frac{c-1}{q} - \sum_{i \neq y_{\boldsymbol{x}}} \mathcal{L}_q(f(\boldsymbol{x}), i))] + \mathbb{E}_D[\sum_{i \neq y_{\boldsymbol{x}}} \eta_{y_{\boldsymbol{x}} i}\mathcal{L}_q(f(\boldsymbol{x}), i)]$$

$$= \frac{c-1}{q}\mathbb{E}_D(1 - \eta_{y_{\boldsymbol{x}}}) - \mathbb{E}_D[\sum_{i \neq y_{\boldsymbol{x}}} (1 - \eta_{y_{\boldsymbol{x}}} - \eta_{y_{\boldsymbol{x}} i})\mathcal{L}_q(f(\boldsymbol{x}), i)],$$

and

$$R^\eta_{\mathcal{L}_q}(f) \geq \frac{c - c^{1-q}}{q}\mathbb{E}_D(1 - \eta_{y_x}) - \mathbb{E}_D[\sum_{i \neq y_x}(1 - \eta_{y_x} - \eta_{y_x i})\mathcal{L}_q(f(x), i)].$$

Hence,

$$(R^\eta_{\mathcal{L}_q}(f^*) - R^\eta_{\mathcal{L}_q}(\hat{f})) \leq \frac{c^{1-q} - 1}{q}\mathbb{E}_D(1 - \eta_{y_x}) +$$
$$\mathbb{E}_D\sum_{i \neq y_x}(1 - \eta_{y_x} - \eta_{y_x i})[\mathcal{L}_q(\hat{f}(x), i) - \mathcal{L}_q(f^*(x), i)].$$

Now, from our assumption that $R_{\mathcal{L}_q}(f^*) = 0$, we have $\mathcal{L}_q(f^*(x), y_x) = 0$. This is only satisfied iff $f_i^*(x) = 1$ when $i = y_x$, and $f_i^*(x) = 0$ if $i \neq y_x$. Hence, $\mathcal{L}_q(f^*(x), i) = 1/q$ $\forall i \neq y_x$. Moreover, by our assumption, we have $(1 - \eta_{y_x} - \eta_{y_x i}) > 0$. As a result, to derive a upper bound for the expression above, we need to maximize the second term. Note that by definition of the $\mathcal{L}_q$ loss, $\mathcal{L}_q(\hat{f}(x), i) \leq 1/q$ $\forall i \in [c]$, and hence the second term is maximized iff $\mathcal{L}_q(\hat{f}(x), i) = 1/q$ $\forall i \neq y_x$. This implies that the maximum of the second term is non-positive, so we have

$$(R^\eta_{\mathcal{L}_q}(f^*) - R^\eta_{\mathcal{L}_q}(\hat{f})) \leq \frac{c^{1-q} - 1}{q}\mathbb{E}_D(1 - \eta_{y_x}).$$

Lastly, since $\hat{f}$ is the minimizer of $R^\eta_{\mathcal{L}_q}(f)$, we have that $R^\eta_{\mathcal{L}_q}(f^*) - R^\eta_{\mathcal{L}_q}(\hat{f}) \geq 0$. This completes the proof. $\qquad\square$

**Lemma 3.** *For any $x$ and $q \in (0, 1)$, assuming $1/c \leq k < 1$ where $c$ represents the number of classes, the sum of truncated $\mathcal{L}_q$ loss with respect to all classes is bounded by:*

$$\tilde{d}k\mathcal{L}_q(\frac{1}{d}) + (c - \tilde{d})\mathcal{L}_q(k) \leq \sum_{j=1}^{c}\mathcal{L}_{trunc}(f(x), e_j) \leq c\mathcal{L}_q(k), \tag{18}$$

*where $\tilde{d} = \max(1, \frac{(1-q)^{1/q}}{k})$.*

*Proof.* For the upper bound, by definition of truncated $\mathcal{L}_q$, $\mathcal{L}_{trunc}(f(x), e_j) \leq \mathcal{L}_q(k)$ for any $x$ and $j$. Hence, $\sum_{j=1}^{c}\mathcal{L}_{trunc}(f(x), e_j) \leq c\mathcal{L}_q(k)$.

For the lower bound, it can be verified that,

$$\sum_{j=1}^{c}\mathcal{L}_{trunc}(\tilde{f}(x), e_j) \leq \sum_{j=1}^{c}\mathcal{L}_{trunc}(f(x), e_j)$$

where $\tilde{f}(x) = (p, \cdots, p, 0, \cdots, 0)$, with $p = 1/d \geq k$ and $d$ is the number of elements in $f(x)$ with a value $\leq k$. Note that since $p > k$, $1 \leq d \leq 1/k$:

$$\sum_{j=1}^{c}\mathcal{L}_{trunc}(\tilde{f}(x), e_j) = d\mathcal{L}_q(p) + (c - d)\mathcal{L}_q(k) = d\mathcal{L}_q(\frac{1}{d}) + (c - d)\mathcal{L}_q(k).$$

We can get a universal lower bound (that does not depend on $f$) by minimizing the above function with respect to $d$. To do so, we treat $d$ to be continuous. By definition of $\mathcal{L}_q$ loss, and recall that $0 < q < 1$,

$$\min_{d \in [1, 1/k]} d\mathcal{L}_q(\frac{1}{d}) + (c - d)\mathcal{L}_q(k) = \min_{d \in [1, 1/k]} d[(1 - (\frac{1}{d})^q)/q - (1 - k^q)/q] = \min_{d \in [1, 1/k]} d[(k^q - (\frac{1}{d})^q)].$$

We can verify using the second derivative test that the above objective function is convex. As a result, we can find the minimum by taking its derivative. Doing so, we find that $d = \frac{(1-q)^{1/q}}{k}$ minimizes the above objective function. Hence, the lower bound is

$$\tilde{d}k\mathcal{L}_q(\frac{1}{d}) + (c - \tilde{d})\mathcal{L}_q(k) \leq \sum_{j=1}^{c}\mathcal{L}_{trunc}(f(x), e_j),$$

where $\tilde{d} = \max(1, \frac{(1-q)^{1/q}}{k})$. $\qquad\square$

**Remark.** *Using Lemma 3, we can prove that the proposed truncated loss leads to more noise robust training following the same arguments as in Theorem 1 and 2.*