[Reviews · NeurIPS 2018]

Reviewer 1



I acknowledge that I read the author's response, and I feel that the revised version of the manuscript will be even stronger. As a result, I am raising my score to an 8. This paper is definitely in the Top 50%. ----- Summary: This paper shows that using the negative Box-Cox transformation as a loss function ($L_q$ or Lq loss) generalizes both categorical cross-entropy (CCE) and mean absolute error (MAE) losses and offers an elegant way to trade off their respective virtues when training on noisy labels. The key insight comes from analyzing the loss function gradients: they are equivalent, except that CCE includes a term that implicitly assigns higher weights to incorrect predictions. This makes training with CCE faster than with MAE but also makes it more susceptible to overfitting label noise. Like CCE, the gradient of Lq loss yields a weighting term but with an exponent parameter that we can choose. When q=0, we get CCE, and when q=1, the weighting term disappears, which is equivalent to MAE. The paper shows that a variant of a known risk bound for MAE under uniform label noise applies to Lq loss as q approaches 1. Experimental results are noticeably strong: Lq consistently outperforms CCE and MAE both and is competitive with several alternative strong baselines. My current feeling is that this paper is a pretty clear accept, but I look forward to reading the other reviewers' comments and authors' responses. Quality: This submission appears sound to me -- the bounds quoted from the Ghosh work are correct and the gradients for the CCE, MAE, and Lq losses are correct. The analyses of Lq loss (lines 147-153) and truncated Lq loss (lines 165-178) appear correct to me, but I can't guarantee their correctness. For what it's worth, I'd recommend including a more detailed proof in either an appendix or an extended version of the paper on arXiv. The experiments provide convincing evidence that Lq loss is superior to CCE and MAE in the presence of uniform and class conditional label noise. Lq loss is competitive with Sukhbaatar and Fergus algorithm, even when it has access to the noise rate. I do have two questions for the authors: (1) How do these results apply to binary classification (presumably they do since binary classification is a special case)? However, Ghosh showed in an earlier paper [4] that there for binary classification, there are other noise robust losses, e.g., ramp loss, that work quite well and might provide a stronger baseline than MAE. (2) Ghosh's results apply to more general label noise scenarios, including non-uniform noise. Does the analysis here also apply to this more challenging (and realistic) scenario? The experiments do have one weakness: there has been a lot of research on noisy labels in the last few years, so I think that a comparison with additional baselines is warranted. Examples include [1][2][3]. Frankly, I'm not sure ANY baseline comparisions is strictly necessary since the paper's central claim is that Lq loss is superior to CCE and MAE -- not that it's superior to all noisy label learning techniques. However, the inclusion of the "Forward" method opens the door to the "why didn't you include this other baselinee" critique. Clarity: This paper is clearly written for the most part, though as noted, the theoretical analysis of Lq loss (lines 147-153, 165-178) would benefit from greater detail, even in a separate document. The analysis of the gradients could use further clarification -- this sentence on lines 122-23 is particularly challenging to understand: "...less congruent with provided labels are implicitly weighed more in the gradient update." Staring at Equation 5, it's not immediately clear to me what this means. The discussion of truncated Lq loss could benefit from further discourse. For example, what is the motivation behind the introduction of the threshold k in Equation 9? How can we intuitively understand what it's doing, and how does it connect to Equation 11? Also, the details of the pruning step aren't entirely clear: since neural net training typically involves unconstrained SGD, how do we optimize v_i bounded between 0 and 1? Originality: This paper has a marginal amount of originality -- it leans heavily on the Ghosh and Ferrari/Yang papers, synthesizing their respective ideas (risk bounds for noise robust symmetric loss functions and Lq loss), rather than proposing entirely new ones. Nonetheless, I think it demonstrates innovtative thinking, especially when looking at the gradients of the loss functions and considering how they will impact the dynamics of neural net learning (wereas Ghosh focuses primarily on the statistical properties). Significance: I'm really uncertain about the significance of the paper. But my takeaway from the paper is "use Lq loss instead of CCE and MAE, in most cases." Given that Lq loss appears pretty easy to implement and differentiate, there seems to be no reason to replace CCE with Lq in any experiment where there may be nontrivial amounts of label noise. References [1] Reed, et al. Training Deep Neural Networks on Noisy Labels with Bootstrapping. ICLR 2015 Workshop Track. [2] Natarajan, et al. Learning with Noisy Labels Nagarajan. ICML 2013. [3] Ma, et al. Dimensionality-Driven Learning with Noisy Labels. ICML 2018. [4] Ghosh, et al. Making Risk Minimization Tolerant to Label Noise. Neurocomputing 2015.

Reviewer 2



Summary A method for learning a classifier robust to label noise is proposed. A new class of robust loss functions is considered to tackle the problem. The contribution is based a truncated and Box-Cox transformed version of the mean absolute loss, which was shown to be theoretically robust to uniform label noise. The proposed method (unlike most previous work) works for both closed and open-set noisy labels. Several experiments support the approach. Detailed comments I liked the insight given in formula (5), which can be considered the starting point of the paper argument. The theoretically-backed derivation of the truncated loss is nice -- in contrast with several heuristics proposed in this area. The straightforward application to open set noise is also interesting. I think this is a solid contribution and deserves publication. A weakness is the need of two new hyperparameters, i.e. k and q. Although, it seems that a default choice worked well throughout all experiments. It would be helpful to know more about the additional computational cost due to the pruning step. How slower is training? Moreover, I think that some details about the proposed alternating optimisation are omitted. How do you guarantee that the v_i are bounded in [0,1] by using SGD? Do you use a proximal method? Minors * line 113: why the fact that CCE is unbounded matter here? And isn’t MAE unbounded as well? * line 143: could you at least show on a footnote the derivation of CCE from Lq when q -> 0 ?

Reviewer 3



Update: after reading the authors' response, I have increased my overall score to "Top 50% of accepted NIPS papers". This paper proposes a new loss function to deal with label noise, which overcomes the drawbacks of standard loss functions (non-robustness to label errors) as well as of existing robust loss functions (they lead to very slow training and a drop in accuracy). Moreover, the proposed family of loss functions contains the cross-entropy loss and the mean absolute error loss as particular (extreme) cases. The paper is well written (although there are many missing determiners, such as "the" or "a") and the ideas are well presented, motivated, and justified. The experiments clearly illustrate the behavior of the proposed loss, confirming that it performs as expected. Important questions remain about the usefulness and applicability of this type of robust loss function. Namely, in the presence of some dataset, how to infer its level of noise (if any) and how to adjust the parameters (q and k) of the loss? If the dataset is noisy (contains wrong labels), arguably cross-validation will not be very informative.